# The Influence of Fly Ash on the Foaming Behavior and Flame Retardancy of Polyurethane Grouting Materials

**DOI:** 10.3390/polym14061113

**Published:** 2022-03-10

**Authors:** Sitong Zhang, Wenying Liu, Kaijie Yang, Wenwen Yu, Fengbo Zhu, Qiang Zheng

**Affiliations:** 1College of Materials Science & Engineering, Taiyuan University of Technology, Taiyuan 030024, China; zhangsitong0356@link.tyut.edu.cn (S.Z.); liuwenying0037@link.tyut.edu.cn (W.L.); yangkaijie0253@link.tyut.edu.cn (K.Y.); zhufengbo@tyut.edu.cn (F.Z.); 2Department of Polymer Science & Engineering, Zhejiang University, Hangzhou 310027, China

**Keywords:** fly ash, grouting, polyurethane, foaming behavior, flame retardance

## Abstract

Polyurethane (PU) grouting material has been widely utilized to control water inrush in mining fields. However, the application has been limited by its high cost and poor flame retardancy. Here, we use the fly ash (FA), a waste from coal of the iron-making industry and power plants, as a partial replacement of conventional filler in PU grouting materials to reduce the production cost and the environmental pollution of FA. The surface-modified FA-filled PU (PU/FA) composites were prepared by room-temperature curing. The effects of FA contents (*φ*) on the structure, foaming behavior, thermal stability, mechanical properties, hydrophobic properties, and flammability of PU grouting materials were examined. Results showed that the higher the *φ*, the more porous the PU/FA composites are, resulting in a lower density and lower mechanical properties. The relationship between the compression modulus *E* and the density *ρ* of the PU/FA composites was *E* ∝ *ρ*^1.3^. In addition, the surface-modified FA improved the compatibility between the hard and soft segment of PU in the PU/FA composite, giving the composites enhanced thermal stability, high hydrophobicity, and flammability resistance.

## 1. Introduction

Owing to the complex underground geological conditions of coal mines, accidents such as coal slippage, gas protrusion, and mine water inrush in fractured rock mass easily occur and seriously affect the safety of underground operators and the efficiency of this process. A widely used and time-proven technique to prevent such mine accidents is to reinforce broken coal and rock by grouting, which can effectively connect and support weak structures. Currently, available grouting materials are generally classified as inorganic materials (cement [1], water glass [2], and clay [3]), organic materials (polyurethane (PU) [4], epoxy resin [5], and acrylamide [6]), and their composites [1,2,3]. Traditional inorganic materials are easy to dilute and hard to coagulate under the influence of water inrush, while the high reactive exotherm, flammability, and high cost of organic materials violate the original purpose of disaster management. In addition, organic materials tend to foam in wet fractures greatly reducing their strength. Therefore, research on grouting materials for coal mines has become an important research direction for the safety of coal mining. For example, a new soluble polymer foaming grouting material was developed by using hydrophilic amino resin as the base material [7], polyurethane/water glass (PU/WG) grouting materials [2], ordinary portland cement (C), high-strength gypsum powder (G), and a small amount of water glass (S) as matrix components to make CGS grouting material [8]. In addition, a modified grouting material was developed by using the double liquid grouting mode [9]. PU is one of the most promising materials and has attracted widespread attention because of its good designability, high permeability, strong adhesion, and easy operation [8,10,11,12]. However, as an organic material, its practical application is limited by its high prices, poor flame retardant properties, poor thermal stability, and other problems. Inorganic filler modification is an important strategy to improve performance and reduce the cost of PU grouting materials. Inorganic substances such as nano-silicon, water glass, red mud, and cement are often used to prepare PU-based organic-inorganic composites [8,12,13,14]. 

Fly ash (FA) is one of the most abundant industrial waste residues in China, and its annual output is approximately 500 million tons [15]. FA is a waste of coal combustion in thermal power plants and consists mainly of silica (SiO_2_), alumina (Al_2_O_3_), iron oxide (Fe_2_O_3_), calcium oxide (CaO), and a small number of other oxides. Its particles are spherical with a diameter ranging from 1–150 microns, depending on the source [15]. Waste FA directly pollutes the water, soil, and atmosphere in the ecological environment. With the accelerated consumption of raw materials, the efficient use of natural resources and the recycling of industrial waste have become increasingly serious. In urban road construction, FA is often used as a reinforcing filling material, such as in FA cement [16], FA lightweight aggregate [17], FA brick [18], and FA concrete [19]. FA is also used as a matrix material in geopolymer production [20]. At present, FA has become one of the main potential filling materials in polymer matrices due to the following advantages: Khoshnoud et al. observed an increase in the energy storage modulus of PVC foam containing FA [21]. Kuźnia et al. reported the improved mechanical property, thermal performance, and heat stability of 20 wt.% FA-filled rigid PU foams [22]. Additionally, Qin et al. stated a positive effect on the crosslinking density and hardness of PU/FA composites, thereby improving their mechanical properties [23]. Qianqian Zhou et al. added FA and intumescent flame retardant (IFR) to the thermoplastic PU (TPU) matrix and found that the synergy between these two materials can improve the flame retardancy of TPU [24]. All these studies on polymer/FA composites confirmed that this filler can improve the mechanical properties, thermal stability, and flame capacity of polymer materials. However, few of them focus on the effect of FA on the foaming behavior and flame retardancy of PU grouting materials.

In this work, FA-filled PU (PU/FA) composites were prepared by simple in-situ curing of the mixture of a surface-modified FA filler and precursor solution with a catalyst at room temperature. The influence of surface-modified FA on the filling morphology of PU/FA composites was investigated by SEM. The effects of FA contents (*φ*) on the foam structure thermal stability, mechanical properties, and flame retardancy of the PU/FA composite foam were also examined. This study is not only helpful for the understanding of the foaming process but also gives a new principle for designing new grouting materials based on FA.

## 2. Materials and Methods

### 2.1. Materials

PM-200 (-NCO% is 30.5–32.0%, industrial grade) was purchased from Wanhua Chemical Group Co., Ltd. (Yantai, China). RT-204 (polyether glycol, hydroxyl number is 320–340 mg KOH/g, the number of functional groups is 2, industrial grade) and RT-305 (polyether polyols based on glycerol, hydroxyl number is 255–310 mg KOH/g, the number of functional groups is 3, industrial grade) were obtained from Nantong Ruitai Chemical Co., Ltd. (Haian, China). Dibutyltin dilaurate (DBTDL) was acquired from Shanghai Adama Reagent Co., Ltd. 3-Amino propyl tri ethoxy silane (APTS) was bought from Nanjing Chuangshi Chemical Additives Co., Ltd. (Leiliu, China), and FA was provided by Shanxi Hujin Coal Power New Materials Co., Ltd. The company analyzed the mean composition of the FA with reference to the GB/T 14563-2020 test standard; FA was mainly composed of SiO_2_ (46.1%), Al_2_O_3_ (40.5%), CaO (2.58%), TiO_2_ (1.91%), and other oxides (8.91%).

### 2.2. Surface Modification of FA

FA was dried at 120 °C for 2 h, then added with APTS, and stirred with a high-speed mixer at 110 °C (Xpdlast SHR-5A) at 1500 r/min for 15 min to obtain a surface-modified FA.

### 2.3. Preparation of PU/FA Composite Foam

According to the formulation shown in Table 1, the corresponding raw materials were weighed. The polyether glycol RT-204 and the diisocyanate RT-305 were labeled as component A, and PM-200 was labeled component B. Firstly, the FA was added to component A in a specific proportion and mixed evenly. Afterward, component B was added to the mixture. Finally, a DBTDL catalyst (0.2% of the total mass) was added and stirred until the reaction became exothermic. Stirring was then terminated, and the solution was injected into a prepared mold. The foam of the PU/FA composite is shown in Figure 1. The surface of the PU grouting material is smooth with a dense interior. Meanwhile, the PU/FA composite material has loose bubble holes in its interior. With the increase in *φ*, the foaming rate of PU/FA composite material increased fourfold because FA retained some of the ammonium sulfate and ammonium thiosulfate from the ammonia flue gas desulfurization [15]. These substances react at high temperatures to generate gas. As a result, the PU/FA composite presents a porous structure.

### 2.4. Testing and Characterization

Fourier transform infrared spectroscopy (FT-IR) was measured using a Fourier infrared spectrometer (Bruker Tensor 27, Coventry, UK). FA powders were prepared using the standard KBr tablet pressing method, and PU and PU/FA composite foams were tested using the attenuated total reflection at a scanning range of 4000–500 cm^−1^.

A field emission scanning electron microscope (ZEISS Gemini 300, Jena, Germany) was used to observe the distribution topography of FA in PU/FA composites. The sample was quenched at a low temperature in liquid nitrogen and the section gold-dusted to improve the conductivity of its surface. The obtained images were analyzed using ImageJ software to determine the foaming rate (%) of the PU/FA composites.

Thermal analysis was performed in a nitrogen atmosphere using a differential scanning calorimeter (TA Instrument DSC Q2000, New Castle, DE, USA) to measure the glass transition temperature of the PU/FA composites. The heating rate was 10 °C/min, the heating range was −70 °C to 250 °C, and the temperature was increased twice.

The thermal stability of the PU/FA composites was tested using a thermogravimetric analyzer (NETZSCH TG-209, Helb, Germany) with a protective atmosphere of nitrogen under the temperature range of 30–700 °C and a heating rate of 10 °C/min.

Tensile and compressive strength was tested in accordance with GB/T 1040-2006 and GB/T 1041-2008 using a universal testing machine (UTM 4304X, Shenzhen, China). The tensile test spline was a dumbbell-type spline and the tensile rate was 5 mm/min. The compression test sample was a cylinder with a diameter of 48 mm; the compression rate was 5 mm/min, and the compression amount was 10% of the length of the specimen. Impact performance was examined in accordance with GB/T 1043-2008 using a pendulum impact tester (XJUG-5.5, Xiamen, China) and an unnotched specimen as the spine. Measurements were expressed as the mean and standard deviation of five trials.

The apparent contact angle was characterized to study the hydrophilic and hydrophobic properties of the PU/FA composites. First, the material surface was wiped clean to eliminate the influence of surface roughness. A water contact angle analyzer (Powereach JC2000D1, Shanghai, China) was used to measure the equilibrium water contact angle of the sample. The sample was then cut and the contact angle of its cross-section was tested. In each measurement, water (10 μL) was dripped onto the sample surface. The average value was calculated from 30 measurements.

Conical calorimetry was performed to test the combustion performance of the PU/FA composites. A cone calorimeter (British FTT-0242 type) was used for the conical calorimetric test in accordance with ISO 5660-1-2002. The sample size was 100 × 100 × 4 mm, and the heat flux was 35 kW/m^2^.

## 3. Results and Discussion

### 3.1. Structural Analysis of FA and PU/FA Composites

FT-IR was used to characterize the structure of FA powders and PU/FA composites. According to the main chemical composition of FA provided by the production company, bands associated with SiO_2_ were well-defined in the spectrum of FA. As shown in Figure 2a, the peak of FA at 1097 cm^−1^ corresponds to the Si-O-Si bond’s stretching vibration, and that at 798 cm^−1^ corresponds to the Si-CH_3_ bond. These results indicated that the peak strength of FA at 1097 and 798 cm^−1^ was enhanced possibly due to the increased or stable presence of Si-O-Si bonds and Si-CH_3_ bonds after the modification by silane coupling agents. This indicates that many polymers adhered to the surface of the FA particles after being modified by a silane coupling agent This phenomenon is consistent with the results reported by T. T. Wu et al. [17]. As shown in Figure 2b, the symmetrical and asymmetrical tensile vibrations of the C-H bonds are reflected at 2933 and 2862 cm^−1^ for the PU and PU/FA composite samples. Additionally, the sharp peaks at 1720 and 1654 cm^−1^ correspond to the C=O tensile vibrations. In all of the PU/FA composites, no signal was detected at 2270 cm^−1^ (-NCO stretch) or 3590 cm^−1^ (-OH stretch), indicating that the final product has no isocyanates and the response was completed.

Figure 3 shows the SEM images for PU, PU/pristine FA-20%, PU/FA-20%, and PU/FA-50%. FA domain is mainly composed of irregular agglomerates with a dispersion size of 1–10 μm. Pristine FA was found to be dispersed homogeneously in the PU matrix. The inorganic-organic interface visible from the PU matrix can be easily detached to form a cavity during quenching. This finding indicated that the interface compatibility is relatively poor, and the composite exhibits poor mechanical properties [22]. For the PU/FA composite, the size of FA particles in the PU/FA composites sample did not change significantly compared with that in the PU/pristine FA composite. Many polymers adhered to the surface of the FA particles after being modified by a silane coupling agent, resulting in no distinct inorganic-organic interface observed. Thus, it can be concluded that the introduction of surfactants is beneficial for the interface compatibility between the FA and PU matrix. When *φ* = 50%, FA remained uniformly distributed and not separated from the PU matrix, as shown in the SEM image. Therefore, modified FA was used to fill PU grouting materials in the subsequent studies.

### 3.2. Effect of FA on the Foaming Behavior of PU/FA Composites

#### 3.2.1. Effect of FA Content

Figure 4 shows the foaming rate, density, and maximum reaction temperature of the PU and PU/FA composites. Due to the large number of combustibles and complex terrain in the coal mines, the excessive local temperature will bring huge hidden dangers to the coal mines [25]. Controlling reaction heat is the most important means to reduce the exothermic disaster caused by PU grouting materials. The curing of PU grouting materials is an exothermic reaction and the maximum reaction temperature of PU/FA composites decreases after FA addition. With the increase in *φ*, the foaming rate of PU/FA composites increased and the density decreased. When *φ* = 50%, the foaming rate of PU/FA-50% was 476.3% and the density was 0.596 g/cm^3^, which was 46.1% lower than that for PU.

Figure 5 shows the SEM images of the PU and PU/FA composites. The interface of PU is relatively smooth and has no bubbles and the interior of the PU/FA composite forms a dense network of holes. In addition, the density increased with *φ*. Image-J software was used to calculate the porosity. The porosities of PU, PU/FA-10%, PU/FA-20%, PU/FA-30%, PU/FA-40%, and PU/FA-50% were 0%, 12.3%, 23.6%, 34.2%, 44.8%, and 48.6%, respectively. With the increase in *φ*, the porosity of PU/FA composites increased and the materials became loose and porous. When *φ* < 20%, the holes closed. When *φ* ≥ 20%, some large size FA particles may damage the structure of the PU matrix, resulting in the opening of the vesicle walls [26]. When FA is evenly distributed in the polymer matrix, its content has minimal effects on vesicle size due to two opposing phenomena, namely, nucleation and cell growth [27]. As a nucleation site, FA particles increase the number of vesicles formed with the increase in filler content; however, the increase in *φ* leads to an increase in the viscosity of the system and hinders the cell growth [28].

#### 3.2.2. Effects of Ambient Humidity

The foaming of PU/FA-20% under different humidity environments at a constant temperature of 18 °C is shown in Figure 6. With the increase in ambient humidity level, the foaming rate of PU/FA composite materials also increased. High humidity increases the moisture content in the environment. When the water in the environment penetrates the interior of the material or forms a water film on its surface, some of the water molecules participate in the curing of PU/FA composites. In addition, humidity accelerates gas emission during the curing of PU/FA composites [29], thus leading to an increase in the foaming rate of PU/FA composites.

### 3.3. Glass Transformation of PU and PU/FA Composites

Figure 7 shows the DSC curve of PU and PU/FA composites. FA addition affected the glass transition temperature *T*_g_ of the PU matrix. The *T*_g_ of PU is 87.31 °C. With the increase in *φ*, the *T*_g_ of PU/FA composite material gradually decreased compared with that of PU. When *φ* = 50%, the *T*_g_ of PU/FA-50% shifts to 56.21 °C, indicating that the compatibility of hard and soft segments has gradually increased. On the one hand, FA addition reduces the binding force of the PU soft segment molecular chain movement and increases the flexibility of the polymer chain [22]. As indicated by the C-H bond and N-H bond in Figure 2b, the strength was significantly reduced; FA addition weakens the hydrogen bonding effect, thereby also weakening the aggregation of PU hard segments [30], indicating that the compatibility of hard and soft segments for PU/FA composites has gradually increased. The widths of the glass transitions, ∆T, as determined by the method shown in Figure 7a, are given as a function of the FA content in Figure 7c. With the increase in *φ*, ∆T also became broad, indicating that FA addition prolongs the glass transition of PU/FA composites.

### 3.4. Thermal Stability of PU and PU/FA Composites

The thermal degradation behavior of FA and PU/FA composites was characterized by thermogravimetric analysis at an N_2_ atmosphere. Figure 8 shows the thermogravimetric (TG) and differential thermogravimetric (DTG) curves. The following thermal parameters calculated by the TG/DTG curve are shown in Table 2: *T*_5%_ represents the initial degradation temperature corresponding to a 5% weight loss; *T*_1_ and *T*_2_ represent the temperature corresponding to the maximum degradation rate of the first and second degradation phases, respectively; and *R*_1_ and *R*_2_ represent the maximum degradation rate of the first and second degradation phases, respectively.

The thermal degradation of PU consisted of two thermal decomposition platforms. The *T*_5%_ of the PU/FA composites was higher than that of PU, indicating that the addition of FA increased the initial thermal degradation temperature of PU and delayed the thermal degradation of PU. When T < 265 °C, the weight loss of PU was small, mainly due to water evaporation and CO_2_ release. The range of 270–400 °C is mainly related to the degradation of PU hard chain segments, including the decomposition of polyols and isocyanate components. As a result, amines, H_2_, CO, CH_4_ are formed [31]. The range of 400–600 °C is mainly the thermal degradation of the PU soft segment. At this temperature, the degradation is slower than that of the hard segment [32].

Table 2 shows that the initial degradation temperature of the PU/FA composites increased with *φ*, while the thermal degradation rates of the PU/FA composites were lower than those of PU within this temperature range. This phenomenon occurred because the barrier effect of FA delays the escape of volatile degradation products from the PU structure [33]. The silica and metal oxides in FA can promote the formation of dense carbon layers, delay the escape of volatile degradation products from the PU structure, and improve the thermal stability of PU in this temperature range. It is noted that *T*_2_ increased significantly with *φ*, indicating an interaction between the soft chain segment of PU and FA, thereby improving the thermal stability.

Thermal stability tests show that the thermal degradation rate of the PU/FA composites slows down and their thermal stability increases. This is consistent with the results obtained in this literature [33]. When *φ* increased to 50%, only one thermal degradation process was found, possibly due to the large number of FA particles that promoted the micro-compatibility of the hard and soft PU segments [34]. This finding is consistent with the DSC results; with the increase in *φ*, the *T*_g_ of PU/FA composites reduced and the glass transition process prolonged, indicating that the compatibility of hard and soft segments for PU/FA composites has gradually increased.

### 3.5. Mechanical Properties of PU/FA Composites

Figure 9 shows the mechanical properties of the PU/FA composites. With the increase in *φ*, the mechanical properties of PU/FA composites followed almost the same law of change as porosity; with a greater porosity, the poorer the mechanical properties and the higher the brittleness. The tensile, impact, and compressive strength of the PU/FA composite decreased with the increase of *φ*. A significant reduction was observed for compression strength at 24.0% when the density of PU/FA decreased from 0.779 g/cm^3^ to 0.596 g/cm^3^. Further study must be conducted on the relationship between the compression modulus E and the density *ρ* of the PU/FA composites. As shown in Figure 9b, the relationship between the E and *ρ* of the PU/FA composites was *E* ∝ *ρ*^1.3^, which is consistent with the relationship between the modulus and density in the classical rigid foam Gibson–Ashby relationship (*E* ∝ *ρ*^1~2^) [35]. For the PU/FA composites with a high FA content, the viscosity of the system increased significantly and further affected their mechanical properties.

### 3.6. Hydrophobic Analysis of PU/FA Composites

Figure 10a shows the apparent contact angle of the cross-section of the PU/FA composites. With the increase in *φ*, the apparent contact angle of the PU/FA composite also gradually increased. Therefore, FA addition improves the hydrophobicity of the PU grouting material. Figure 10b,c shows the apparent contact angles of PU and PU/FA-50%. When *φ* > 40%, the porosity of the PU/FA composite material became extremely large, and the water droplets were absorbed by the pores on the solid surface. As a result, the hydrophobicity of PU/FA-50% decreased.

SEM showed that FA particles are evenly distributed in the PU grouting material. The PU/FA composites have a rough morphology and therefore have better hydrophobicity and water resistance than PU composites with a smooth cross-section [36]. The hydrophilic and hydrophobic properties of grouting materials are rarely mentioned in the literature.

### 3.7. Flame Retardant Properties of PU/FA Composites

The heat release rate (HRR) and total heat release (THR) curves of PU/FA composites are shown in Figure 11a,b, and the smoke release rate (SPR) and total smoke generation (TSP) curves are shown in Figure 11c,d. Additional relevant data are listed in Table 3, including the flameout time (TTF), the average heat release rate (AHRR), THR, and TSP. As shown in Table 3, the TTF of the PU/FA composites was smaller than that of the PU grouting materials, indicating that FA filling reduces the combustion time of PU/FA composites. The TTF of PU/FA-50% was 55.75% lower than that of the PU grouting materials.

As shown in Figure 11a,b, and Table 3, the AHRR of the PU grouting material was 87.16 kW/m^2^ and its THR within 1300 s was 22.67 MJ/m^2^. After FA filling, the AHRR and THR of the PU/FA composites gradually decreased with the increase in *φ*. When *φ* = 50%, the AHRR (53.88 kW/m^2^) and THR (14.01 MJ/m^2^) of the PU/FA composites reached the lowest values for the entire formulation system, implying that the best flame retardant effect was achieved. As shown in Figure 11a, the HRR curves of the PU grouting material, PU/FA-10%, and PU/FA-20% all have three peaks. The FA made the second HRR peak weakened and earlier. When *φ* = 30%, the second peak disappeared and the HRR curve split into two peaks. Similarly, the third peak decreased with the increase in *φ*. When *φ* = 50%, the third peak almost merged with the first peak. This phenomenon indicates a gradual increase in the compatibility of the hard and soft segments of the PU substrates, which is consistent with the conclusions reached by DSC and TG.

As shown in Figure 11c,d, and Table 3, FA addition significantly reduced the SPR and TSP of the PU/FA composites. As shown in Table 3, the TSP of the PU grouting material was 14.31 m^2^, and that of the PU/FA composite decreased with the increase in *φ*. When *φ* = 50%, the TSP was 31.73% lower than that of the PU grouting material, indicating that FA filling can inhibit the smoke release of the PU grouting material. On the one hand, the main components of FA are thermally stable oxides that adhere uniformly to the PU grouting material matrix, playing an insulating role, blocking the diffusion of heat and gas, and preventing pyrolysis products from overflowing [37]. On the other hand, the foaming structure of the PU/FA composite itself provides good adsorption and reduces the smoke generation rate [38].

The results of the combustion test show that the PU/FA composites exhibit good flame retardancy and smoke suppression. This is consistent with the conclusions obtained in the literature [38]. When *φ* = 50%, the AHRR and THR of the PU/FA composites reached the lowest values for the entire formulation system, implying that the best flame retardant effect was achieved. HRR and SPR have only one peak, this phenomenon indicates a gradual increase in the compatibility of the hard and soft segments of PU substrates, which is consistent with the conclusions reached by DSC and TG.

## 4. Conclusions

The effects of FA contents (*φ*) on the structure, thermal stability, hydrophobic properties, mechanical properties, and combustion properties of PU/FA composites were further studied. The modified FA and PU matrix is tightly bound without visible interface separation. The addition of FA reduces the maximum reaction temperature of the PU/FA composites. The foaming behavior of the PU/FA composites was significantly affected by FA filling amount and humidity. With the increase in *φ*, the *T*_g_ of PU/FA composites reduced and the glass transition process prolonged, indicating that the compatibility of hard and soft segments was caused by the weakened H-bond interaction within the hard segment. It follows that the thermal degradation rate of the PU/FA composites slows down and the thermal stability increases. Given that the final mechanical properties of a material are affected by its density, the mechanical properties of the PU/FA composites can be greatly adjusted by changing *φ*. The relation between the compression modulus and density of the PU/FA composites is *E* ∝ *ρ*^1.3^. A modest increase in *φ* also increases the hydrophobic properties of the PU/FA composites. During combustion, the TTF, AHRR, THR, and TSP of the PU/FA composites decreased, the TTF, THR, and TSP of PU/FA-50% were 55.75%, 38.20%, and 31.73% lower than those of PU grouting materials, respectively. Therefore, PU/FA composites exhibit good flame retardancy and smoke suppression.

## Figures and Tables

**Figure 1 polymers-14-01113-f001:**
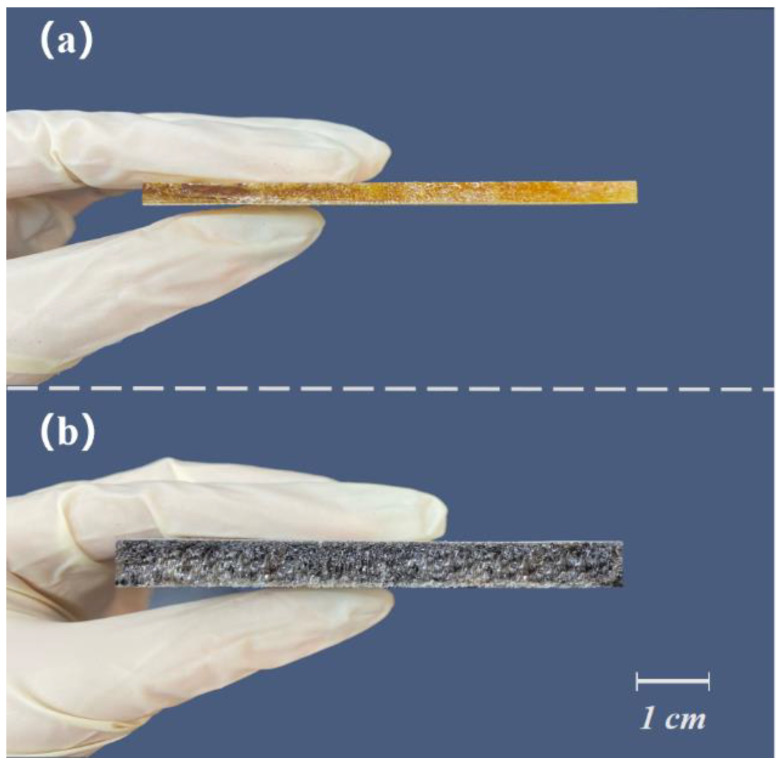
A sample section of PU (**a**) and PU/FA composites (**b**).

**Figure 2 polymers-14-01113-f002:**
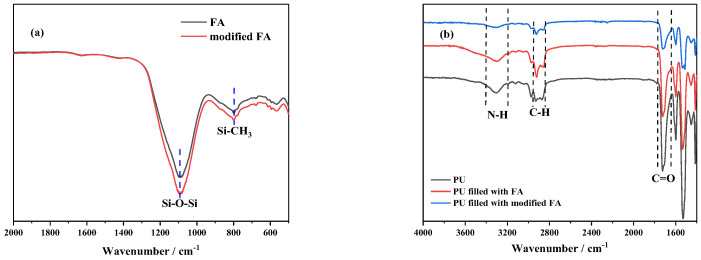
Fourier transform infrared spectroscopy (FT-IR) analysis of FA and modified FA (**a**); PU, PU filled with FA and PU filled with modified FA (**b**).

**Figure 3 polymers-14-01113-f003:**
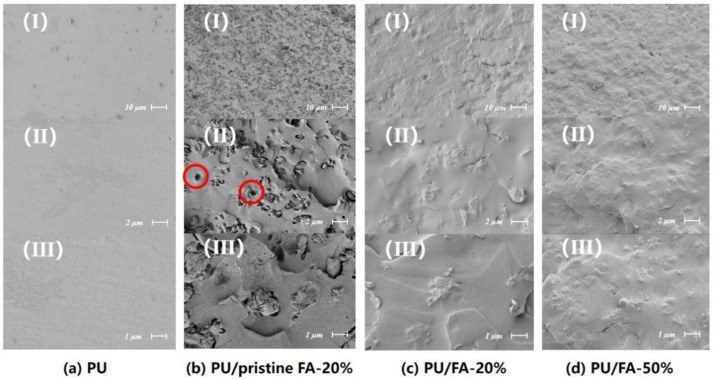
Scanning electron microscopy (SEM) analysis of PU and PU/FA composites.

**Figure 4 polymers-14-01113-f004:**
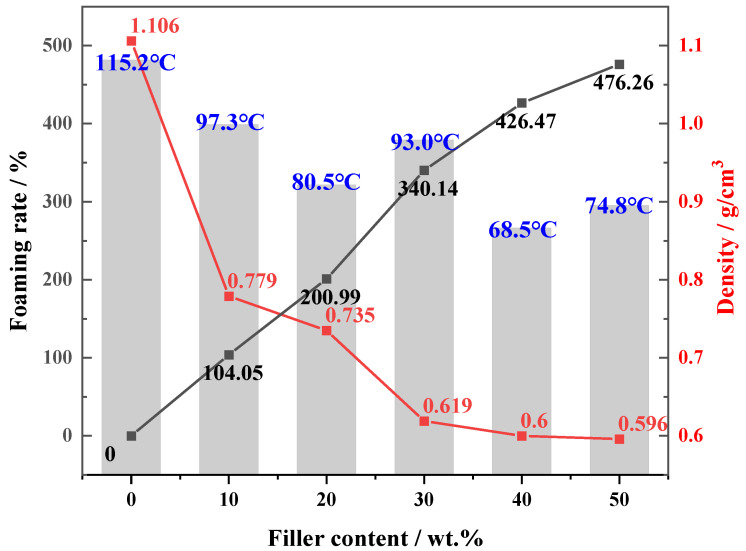
The foaming rate, density, and maximum reaction temperature of the PU and PU/FA composite.

**Figure 5 polymers-14-01113-f005:**
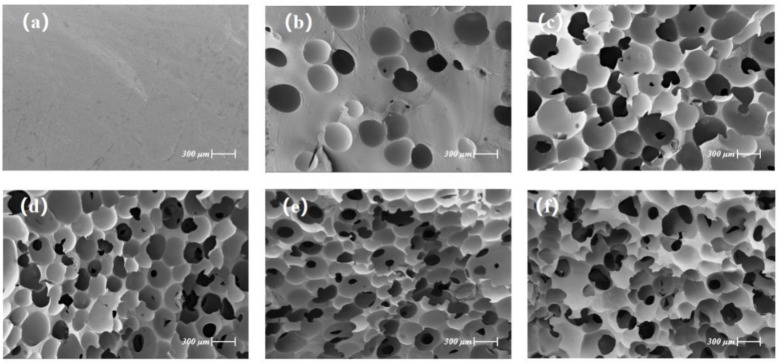
Scanning electron microscopy (SEM) analysis of PU and PU/FA composites: (**a**) PU; (**b**) PU/FA-10%; (**c**) PU/FA-20%; (**d**) PU/FA-30%; (**e**) PU/FA-40%; and (**f**) PU/FA-50%.

**Figure 6 polymers-14-01113-f006:**
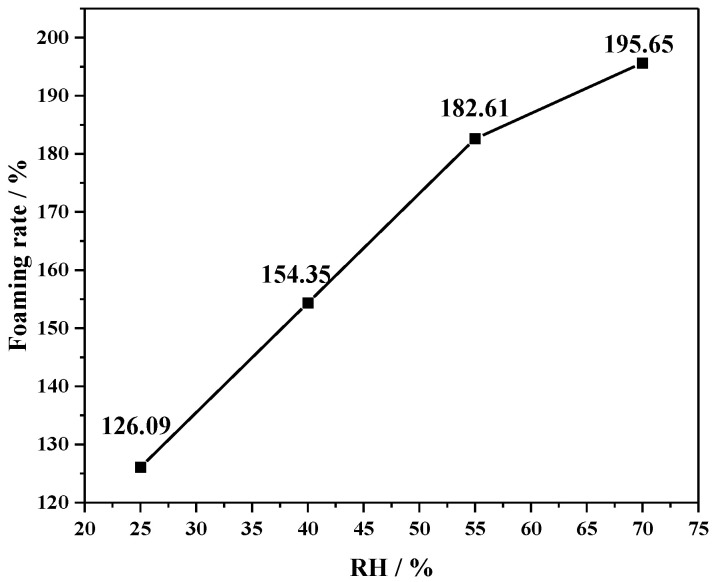
The foaming rate of PU/FA-20% under different humidity environments.

**Figure 7 polymers-14-01113-f007:**
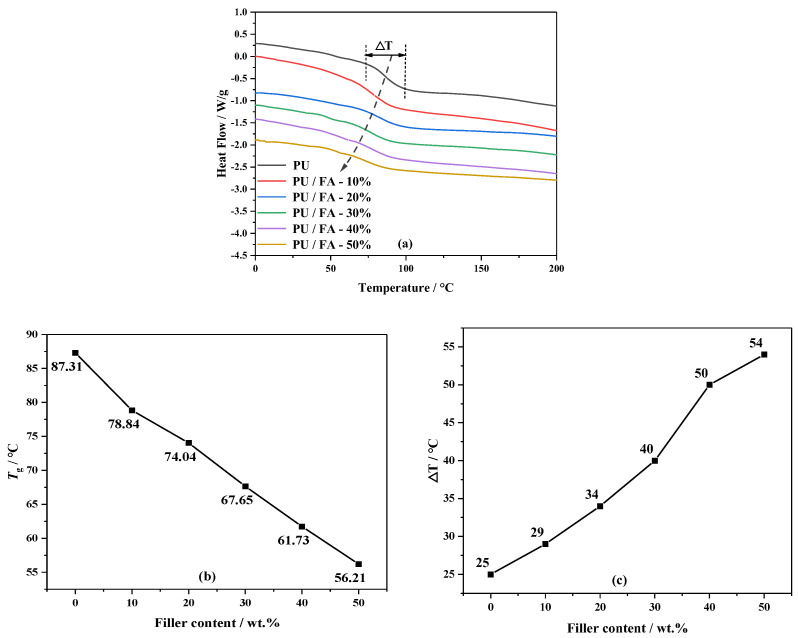
The DSC curve (**a**); *T*_g_ (**b**); ∆T (**c**) of PU and PU/FA composites.

**Figure 8 polymers-14-01113-f008:**
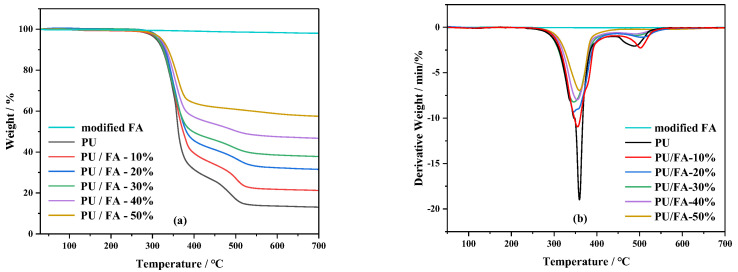
The TG curve (**a**) and DTG curve (**b**) of modified FA, PU, and PU/FA composites.

**Figure 9 polymers-14-01113-f009:**
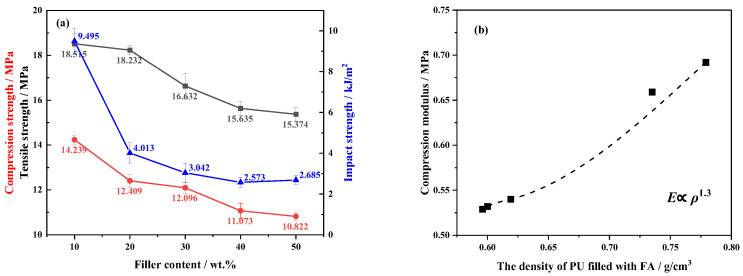
(**a**) The mechanical properties of PU/FA composites; (**b**) compression modulus as a function of density.

**Figure 10 polymers-14-01113-f010:**
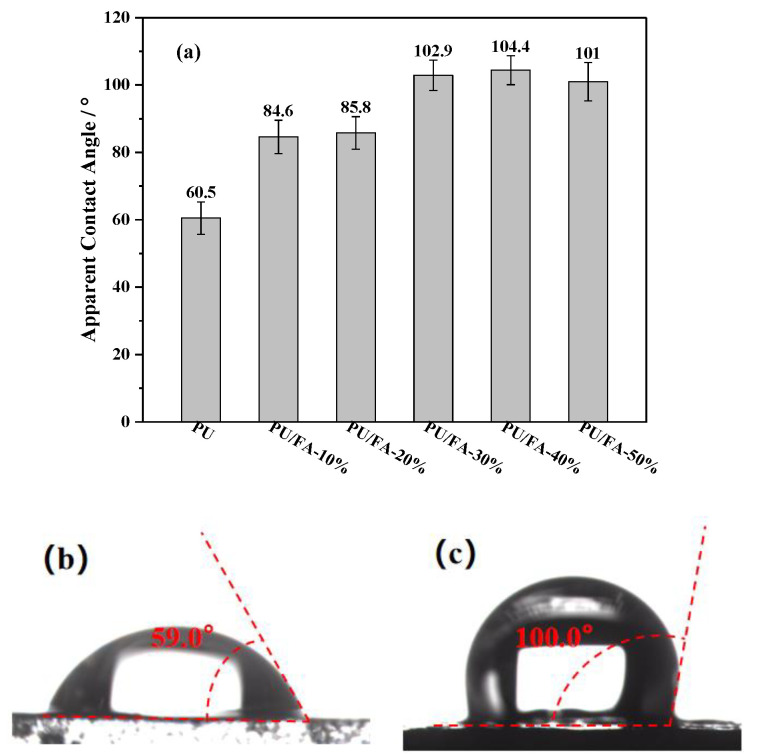
(**a**) PU/FA composites apparent contact angle and apparent contact angle photos: (**b**) PU; (**c**) PU/FA-50%.

**Figure 11 polymers-14-01113-f011:**
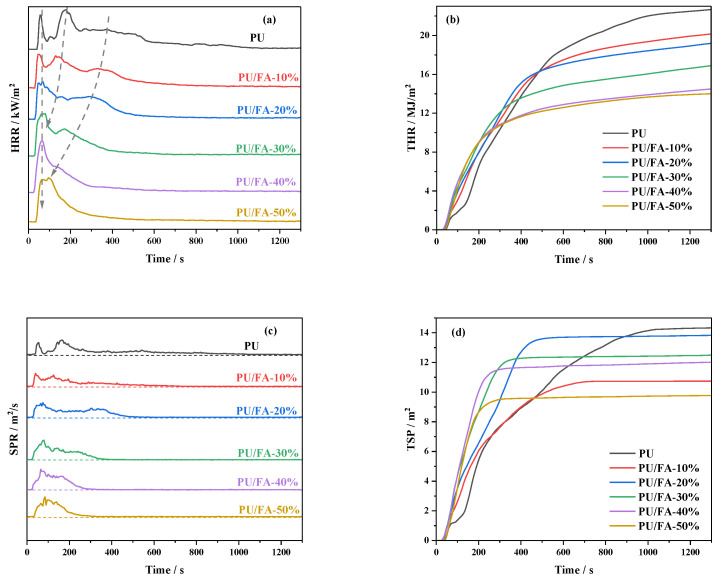
Combustion performance of PU and PU/FA composites. (**a**) HRR curve; (**b**) THR curve; (**c**) SPR curve; and (**d**) TSP curve.

**Table 1 polymers-14-01113-t001:** Formulations of PU and PU/FA composites.

Sample Name	RT-204/%	RT-305/%	PM-200/%	FA/%
PU	16.9	30.1	53.0	0
PU/FA-10%	15.2	27.1	47.7	10
PU/pristine FA-20%	13.5	24.1	42.4	20
PU/FA-20%	13.5	24.1	42.4	20
PU/FA-30%	11.8	21.1	37.1	30
PU/FA-40%	10.1	18.1	31.8	40
PU/FA-50%	8.4	15.1	26.5	50

**Table 2 polymers-14-01113-t002:** The following thermal stability parameters are calculated by the TG/DTG curve for the PU and PU/FA composites.

Sample Name	*T*_5%_/°C	*T*_1_/°C	*R*_1_/%/Max	*T*_2_/°C	*R*_2_/%/Max
PU	312.32 ± 2.16	346.40 ± 9.22	−25.016 ± 4.53	484.03 ± 5.89	−2.103 ± 0.24
PU/FA-10%	310.54 ± 2.70	347.85 ± 5.01	−15.761 ± 4.96	483.20 ± 1.90	−2.069 ± 0.18
PU/FA-20%	313.60 ± 3.57	344.99 ± 2.55	−10.435 ± 0.89	487.02 ± 3.62	−1.662 ± 0.41
PU/FA-30%	313.51 ± 1.03	346.58 ± 2.14	−8.535 ± 0.49	479.79 ± 7.62	−1.444 ± 0.40
PU/FA-40%	318.91 ± 1.08	349.47 ± 3.39	−7.468 ± 0.38	478.76 ± 9.11	−1.118 ± 0.28
PU/FA-50%	320.74 ± 2.41	353.59 ± 4.14	−7.423 ± 0.33		

**Table 3 polymers-14-01113-t003:** Cone test data of PU and PU/FA composites.

Sample Name	TTF/s	AHRR/kW/m^2^	THR/MJ/m^2^	TSP/m^2^
PU	1087	87.16	22.67	14.31
PU/FA-10%	790	77.49	20.15	10.71
PU/FA-20%	660	73.79	19.19	13.81
PU/FA-30%	605	64.96	16.90	12.48
PU/FA-40%	477	55.72	14.49	12.01
PU/FA-50%	481	53.88	14.01	9.77

## Data Availability

The data that support the findings of this work are available from the corresponding author on reasonable request.

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
