# Peer review of "The Influence of Fly Ash on the Foaming Behavior and Flame Retardancy of Polyurethane Grouting Materials"

_polymers, 2022, doi:10.3390/polym14061113_

Round 1

Reviewer 1 Report

The manuscript under the title: “The influence of fly ash on the foaming behavior and flame retardancy of polyurethane grouting materials” is in line with the Polymer journal. This article is based on original research. The research in the article is well presented, but the work is not complete. There is a lack of discussion part and comparison obtained results with up-to-date literature. Additionally, some other  content requires to be improvement, including:

  • Abstract: please clarify main aim.
  • Abstract (line 20): compared – use small letter.
  • Introduction (line 30): please specify the content of each reference [1-6]. The given references should be connected with specific presented topic and each of them should give a new information for the presented background.
  • Introduction (line 30): please also specify the application in geopolymer production, for example: https://doi.org/10.3390/ma14020400
  • Introduction: please define clearly the gap in the literature that justified selected topic and stress the novelty of provided research (in last paragraph to Introduction part).
  • Materials and methods: (point 2.1.) Could you specify the type of FA (chemical composition)? This information may also be gained from the supplier.
  • Figure 1 or in the related text. Could you define the size of the samples?
  • Figure 3 and 7 – please enlarge this figures.
  • Results (line 242): apply down index.
  • Discussion: the discussion part and comparison with up-to-date literature is required.

Reviewer 2 Report

The paper "The influence of fly ash on the foaming behavior and flame retardancy of polyurethane grouting materials" may be publishable after the major, mandatory revision.

Remarks:

  1. Figures 1, 3, 4, 10 need clear scale bars.
  2. Statistical scattering of the thermal stability parameters reported in Table 2 should be supplied under the revision.

3. The notion of the "water contact angle" used by the authors is senseless and misleading, the notion of the "apparent contact angle", which is the only measurable equilibrium contact angle, should be used, see:

Drelich J. et al. Contact angles and wettability: towards common and accurate terminology, Surface Innovations, 5 (2017) 3-8.

Bormashenko Ed. Physics of Wetting. Phenomena and Applications of Fluids on Surfaces, de Gruyter, Berlin, 2017.

The use of the correct, modern scientific wording is of a primary importance.

4. Apparent contact angles demonstrate usually high statistical scattering due to the phenomenon of the contact angle hysteresis. The statistical scattering  is not reported in the paper and definitely should be discussed under the revision.

5. Conclusions Section:

in the text: "The effects of φ on the structure, thermal stability, hydrophobic properties, mechanical properties, and combustion properties of PU/FA composites were further studied".

The precise meaning of the parameter φ should be supplied in the Conclusions Sections.

Round 2

Reviewer 1 Report

The revised manuscript under the title: “The influence of fly ash on the foaming behavior and flame retardancy of polyurethane grouting materials” is significantlly improve. Before the publication are recommended two slight changes: a) editing works must be done b) the point 3.1. about structural research should be commented more detailed. 

Reviewer 2 Report

The revised paper is publishable.

Author Response

Thank you very much for your suggestions and requests.